# Serum S100B Levels in Patients with Liver Cirrhosis and Hepatic Encephalopathy

**DOI:** 10.3390/diagnostics13030333

**Published:** 2023-01-17

**Authors:** Mo-Jong Kim, Jung-Hee Kim, Jang-Han Jung, Sung-Eun Kim, Hyoung-Su Kim, Myoung-Kuk Jang, Sang-Hoon Park, Myung-Seok Lee, Ki Tae Suk, Dong Joon Kim, Eun-Kyoung Choi, Ji-Won Park

**Affiliations:** 1Ilsong Institute of Life Science, Hallym University, Seoul 07247, Republic of Korea; 2Department of Biomedical Gerontology, Graduate School of Hallym University, Chuncheon-si 24252, Republic of Korea; 3Department of Internal Medicine, Dongtan Sacred Heart Hospital of Hallym University Medical Center, 7, Keunjaebong-gil, Hwaseong-si 18450, Republic of Korea; 4Institute for Liver and Digestive Diseases, Hallym University, Chuncheon-si 24252, Republic of Korea; 5Department of Internal Medicine, Hallym University Sacred Heart Hospital of Hallym University Medical Center, 22, Gwanpyeong-ro 170 beon-gil, Dongan-gu, Anyang-si 14068, Republic of Korea; 6Department of Internal Medicine, Kangdong Sacred Heart Hospital of Hallym University Medical Center, 18, Cheonho-daero 173-gil, Gangdong-gu, Seoul 05355, Republic of Korea; 7Department of Internal Medicine, Kangnam Sacred Heart Hospital of Hallym University Medical Center, 1, Singil-ro, Yeongdeungpo-gu, Seoul 07441, Republic of Korea; 8Department of Internal Medicine, Chuncheon Sacred Heart Hospital of Hallym University Medical Center, 77, Sakju-ro, Chuncheon-si 24253, Republic of Korea

**Keywords:** S100B, hepatic encephalopathy, MELD, liver cirrhosis

## Abstract

Hepatic encephalopathy (HE) is one of the main complications of liver cirrhosis (LC) and is classified into minimal hepatic encephalopathy (MHE) and overt hepatic encephalopathy (overt HE). S100B is expressed mainly in astrocytes and other glial cells, and S100B has been reported to be associated with various neurological disorders. The present study aimed to investigate the diagnostic ability of serum S100B to discriminate the grade of HE and the parameters correlated with serum S100B levels. Additionally, we investigated whether serum S100B levels can be used to predict 1-year mortality in cirrhotic patients. In total, 95 cirrhotic patients were consecutively enrolled and divided into the following three groups: (i) without any types of HEs; (ii) with MHE; and (iii) with overt HE. The diagnosis of MHE was made by the Mini-Mental State Examination (MMSE) and Psychometric Hepatic Encephalopathy Score (PHES). Among the three groups, there were no significant differences in serum S100B levels regardless of HE severity. The clinical parameters correlated with serum S100B levels were age, serum bilirubin, and creatinine levels. The Model for End-Stage Liver Disease (MELD) score showed a significant positive correlation with serum S100B levels. The relationship between serum S100B levels and MELD score was maintained in 48 patients without any type of HE. Additionally, hyperammonemia, low cholesterol levels, and the combination of serum S100B levels ≥ 35 pg/mL with MELD score ≥ 13 were factors for predicting 1- year mortality. In conclusion, serum S100B level was not useful for differentiating the severity of HE. However, we found that serum S100B levels can be affected by age, serum bilirubin, and creatinine in cirrhotic patients and are associated with MELD scores. Additionally, serum S100B levels showed the possibility of predicting 1-year mortality in cirrhotic patients. These findings suggest that serum S100B levels may reflect liver dysfunction and prognosis in liver disease.

## 1. Introduction

S100B is a protein belonging to the S100 protein family that was first reported by B.W. Moor in 1965 [1]. The S100 proteins, a family of calcium (Ca^2+^)-binding cytosolic proteins, are composed of 25 known members, namely, S100A1-18, hair hyaluronan, keratin fibrin, reptin, S100B, S100P, S100Z and S100G [2]. These proteins have diverse intracellular and/or extracellular functions, including the regulation of cell proliferation, differentiation, apoptosis, migration, energy metabolism, calcium balance, protein phosphorylation, and inflammation [3,4,5,6]; therefore, altered expression of S100 proteins may be associated with disease development. S100B has been reported to be expressed mainly in astrocytes and other glial cells, such as oligodendrocytes and Schwann cells. It has been reported to be associated with various neurological disorders [7]. However, S100B has also been detected in definite nonneural cell types, such as dendritic cells, certain lymphocyte subpopulations, chondrocytes, Langerhans cells, melanocytes, adrenal medulla satellite cells, skeletal muscle satellite cells, and adipocytes [8,9,10,11]. Additionally, we have previously reported that S100B is expressed in the liver of mice and that the interaction with the receptor for advanced glycation end products (RAGE) is observed in cholestatic liver injury by bile duct ligation [12]. Another experimental study has reported that serum S100B levels increase in the setting of local ischemia and perfusion of the rat liver without intracranial injury, suggesting that the liver may be a possible source of S100B [13].

To date, serum levels of S100B protein have been extensively studied in certain conditions, such as neural tissue injury, but rarely in relatively stable cirrhotic patients without any types of hepatic encephalopathy (HE). LC is the end stage of various liver diseases and a major cause of morbidity and mortality worldwide. HE is one of the main complications of LC and develops in 30–45% of patients with cirrhosis [14]. HE is defined as brain dysfunction caused by liver insufficiency and/or portosystemic shunting. Its manifestations include a wide spectrum of neurological or psychiatric abnormalities, such as inappropriate behavior, disorientation, confusion, slurred speech, stupor, and coma [15]. Depending on the severity of symptoms, HE can be classified into minimal hepatic encephalopathy (MHE) and overt hepatic encephalopathy (overt HE). Several studies have suggested that elevated serum S100B levels may be a diagnostic marker of overt HE in fulminant hepatic failure or cirrhosis [16,17]. However, little is known about the roles of serum S100B levels in diagnosing MHE or predicting 1-year mortality in cirrhotic patients.

In the present study, we aimed to investigate the diagnostic ability of serum S100B to discriminate MHE and the parameters correlated with serum S100B levels in cirrhotic patients, depending on whether HE is present. Second, we investigated whether serum S100B levels can be used to predict 1-year mortality in cirrhotic patients.

## 2. Materials and Methods

### 2.1. Patients

This prospective observational study included all consecutive adult patients with LC admitted to the Hallym University Sacred Heart Hospital in the Republic of Korea from July 2013 to July 2014. The diagnosis of LC was made by clinical, laboratory, and radiologic features. Patients with a history of malignancy or brain diseases, such as stroke, trauma, or psychiatric disease, were excluded. A total of 95 patients were enrolled, and all patients were divided into the following three groups: (i) without any types of HEs; (ii) with MHE; and (iii) with overt HE. To assess the severity of HE, the West Haven criteria were used (Table 1).

The diagnosis of MHE was made according to the Mini-Mental State Examination (MMSE) and Psychometric Hepatic Encephalopathy Score (PHES). All subjects participated in this study after obtaining written informed consent. The Institutional Review Board at the Hallym University Sacred Heart Hospital approved this study (2013–I103).

### 2.2. Diagnosis of MHE and Overt HE

Patients with marked changes in consciousness and behavior corresponding to grades 3–4 of the Western Haven criteria were diagnosed as overt HE. Patients classified as grade 0–2 of the Western Haven criteria underwent MMSE. When the MMSE score was 24 or more, PHES was additionally performed to check for MHE. The PHES was calculated based on the results of the following five paper-and-pencil tests: digit symbol test (DST), number connection test A (NCT-A), number connection test-B (NCT-B), serial dotting test (SDT), and line tracing test (LTT). MHE was diagnosed when the PHES was −5 or less, and patients who satisfied both MMSE score ≥ 24 and PHES > −5 were demonstrated to be free of HE (Figure 1).

### 2.3. Measurement of Serum S100B Levels

Blood samples for analysis of S100B were collected in EDTA tubes, and the plasma was separated and stored at −80 °C until analysis. Serum S100B protein levels were measured using a commercially available enzyme-linked immunosorbent assay (Biovendor, LLC, Chandler, NC, USA). Serum samples were added to the wells of the ELISA plate and incubated. The plates were then washed and incubated with avidin-conjugated horseradish peroxidase-conjugated substrate solution, and HCL (1.0 M) was added to stop the reaction. The optical density (OD) was measured spectrophotometrically at 490 nm on a microplate reader. All serum samples were tested in duplicate, and the average was reported. The minimum detectable concentration was 15 pg/mL. The intra- and interassay precision values were 3.3% and 7.7%, respectively. The serum S100B assays were conducted at the Ilsong Institute of Life Science at Hallym University. The assays were performed by one technician who was blinded to the study groups. 

### 2.4. Statistical Analysis

Statistical analyses were performed using SPSS version 24 (IBM Corp, Armonk, NY, USA). Quantitative variables are expressed as the mean ± standard deviation (SD). ANOVA with a Scheffe post hoc test was used to compare three groups. Correlations between biochemical factors and S100B were evaluated using Pearson’s correlation. A *p* value less than 0.05 was considered significant, and all *p* values were two-tailed. Predictive factors for 1-year mortality were analyzed with uni- and multivariate logistic regression models. Variables with *p* < 0.05 in the univariate model were entered into the multivariate stepwise logistic regression model (forward selection and likelihood ratio) to assess the independent predictive effect of the variables of interest.

## 3. Results

### 3.1. Patient Characteristics

During the study period, 95 patients with LC were prospectively included. The mean age of the participants was 52.77 ± 10.23 years, and 70 (73.7%) patients were male. The etiologies of LC included viral (n = 24), alcoholic (n = 63), and other causes (n = 8). The severity of liver dysfunction was described by the Model for End-Stage Liver Disease (MELD) and the Child-Turcotte-Pugh (CTP) classification. In 22 patients (23.2%), MHE was diagnosed, and overt HE was observed in 25 (26.3%) patients. Moreover, 48 patients were classified as being free of HE. The baseline clinical characteristics of the three groups are shown in Table 2.

### 3.2. Serum S100B and Ammonia Levels According to the Degree of HE

Because S100B has been demonstrated to be a potential marker of brain injury, we evaluated the correlation between serum S100B levels and the degree of HE. The serum S100B levels were 38.1 ± 17.2, 36.5 ± 13.9, and 43.5 ± 19.7 pg/mL in the non-HE, MHE, and overt HE groups, respectively (Table 2). In patients with overt HE, serum S100B levels were slightly high, but there were no significant differences among the three groups. Serum ammonia levels were a better predictor for overt HE than S100B (Figure 2). Serum ammonia levels were significantly higher in the overt HE group than in the non-HE group; however, serum ammonia level could not discriminate patients with MHE from the non-HE group or the overt HE group.

### 3.3. Parameters Correlated with Serum S100B Levels

To identify clinical parameters associated with increased S100B levels, patients were divided into three groups according to serum S100B levels: <25 pg/mL, 25~<34 pg/mL, and ≥34 pg/mL. The group with serum S100B ≥34 pg/mL was characterized as follows: older; higher levels of bilirubin, creatinine, and MELD score; lower levels of platelet; and relatively fewer males (Table 3).

The relationships of serum S100B levels with clinical and biochemical parameters are shown in Table 4. Serum total cholesterol, platelet, and albumin levels negatively correlated with serum S100B levels. In contrast, age, serum bilirubin, uric acid, lactate dehydrogenase (LD), alanine aminotransferase (ALT), INR, and creatinine levels showed a positive correlation with serum S100B levels. Among these, serum creatinine, bilirubin, and INR are the main parameters of the MELD scoring formula. For this reason, it was presumed that a significant relationship was observed between the MELD score and serum S100B levels.

To avoid the multicollinearity between individual parameters (i.e., INR, bilirubin, and creatinine) and the MELD score, the multivariate linear regression analysis was performed, excepting the MELD score (Table 5). Age, serum bilirubin, and creatinine levels were revealed to be significantly related to serum S100B levels.

### 3.4. Serum S100B Level and 1-Year Mortality

Among 95 patients, 21 patients (22.1%) died after 1 year. To identify 1-year mortality-associated factors, clinical characteristics were compared between surviving (n = 74) and non-surviving patients (n = 21) (Table 6). The non-surviving patients were characterized as follows: older; higher levels of S100B, ammonia, white blood cell (WBC), bilirubin, and INR; and lower levels of cholesterol and albumin. The MELD and CTP scores were significantly higher in the non-surviving group compared to the surviving group. There was no significant difference in the distribution of HE between the two groups. 

Multivariate analysis demonstrated that high ammonia levels, low cholesterol levels, and a combination of high MELD score and serum S100B levels were independently associated with 1-year mortality (Table 7). Because S100B showed a higher association with MELD than the severity of HE, we investigated whether S100B contributes to predicting 1-year mortality. Using the ROC curve, the cutoff values of the MELD score and serum S100B level related to 1-year mortality were determined to be 13 and 35 pg/mL, respectively. We divided the patients into the following four groups: (i) S100B < 35 pg/mL and MELD score < 13; (ii) S100B ≥ 35 pg/mL and MELD score < 13; (iii) S100B < 35 pg/mL and MELD score ≥ 13; and (iv) S100B ≥ 35 pg/mL and MELD score ≥ 13. The Kaplan–Meier survival curve showed significantly higher 1-year mortality rates when the S100B level and MELD score were higher than the cutoff value (Figure 3A). To clarify the predictive power of S100B for 1-year mortality, we divided study subjects into three groups depending on the levels of serum S100B (<25 pg/mL, 25 ≤ S100B < 35 pg/mL, and 35 pg/mL ≤). The higher the S100B levels, the higher mortality rates were observed (Figure 3B).

### 3.5. Clinical Significance of Serum S100B in the Non-HE Group

To clarify the clinical significance of serum S100B levels, we investigated the relationship between serum S100B and biochemical parameters in 48 patients without any types of HE. Total bilirubin and the MELD score showed a positive correlation with serum S100B levels (Table 8). Among the 48 patients, seven patients (14.6%) died during 1 year. Interestingly, a significant portion of non-surviving patients (71.4%) showed high serum S100B levels and MELD scores above the cutoff value previously mentioned (Figure 4). However, when S100B (≥35 pg/mL) or MELD score (≥13) was applied separately, there was no statistical significance.

## 4. Discussion

In the present study, serum S100B level was not useful for differentiating the severity of HE. However, serum S100B levels were associated with age and the main parameters of the MELD score, such as serum bilirubin and creatinine levels in cirrhotic patients. Additionally, serum S100B levels showed the potential in the prediction of 1-year mortality in cirrhotic patients.

LC is a life-threatening disorder leading to 1.03 million deaths per year worldwide [18]. Patients with decompensated LC characterized by variceal bleeding or ascites have a 10-fold higher risk of death than the general population [19]. The CTP and MELD scores have been widely used as noninvasive tools for predicting the prognosis of LC. Because the CTP scoring system includes rather subjective variables, such as the degree of evaluation of ascites and HE, the MELD score is considered a more reproducible and accurate tool than the CTP score. In the present study, serum S100B levels showed a positive correlation with MELD scores, and the relationship was more distinct than that with CTP scores.

The biological half-life of S100B has been reported to be in the range of 30–132 min, depending on the clinical situation [20,21,22]. Similar to other small proteins, S100B is presumed to be eliminated by degradation in the proximal tubules in the kidney. Therefore, impaired renal function may affect the serum levels of S100B. Although a significant difference in renal function was not observed among our study groups, the serum S100B level was correlated well with the serum creatinine level, which is one of the main parameters of the MELD scoring system. Regarding the elimination of S100B, Jönsson et al. reported that S100B levels were not influenced by a moderate decrease in glomerular filtration rate (GFR) [22]. At the same time, Gross et al. demonstrated that serum S100B was negatively related to creatinine clearance [23]. In cirrhotic patients, the serum creatinine level is not an accurate marker for estimating renal function due to various factors, such as reduced muscle mass, increased tubular secretion of creatinine, and impaired production of creatine, which is the precursor of serum creatinine. Therefore, to establish the relationship between S100B levels and renal dysfunction in cirrhotic patients, further study using other renal biomarkers, such as cystatin C, is necessary.

In the present study, total serum bilirubin level was associated with serum S100B levels. Similar to our result, Okumus et al. found a significant correlation between serum bilirubin and S100B levels [24]. However, it is unclear how serum bilirubin levels affect serum S100B levels. Serum bilirubin levels have been considered an important factor in the diagnosis and prognosis of patients with liver disorders. Almost all prognostic scores for liver diseases include bilirubin in their calculations. These findings suggest that liver dysfunction can affect serum S100B levels. Additionally, the present study showed that age was associated with serum S100B levels. Similar to our data, Gross et al. found that serum S100B was positively related to age [23]. On the other hand, Portela et al. demonstrated that a correlation between age and S100B was not evident in individuals >20 years of age but was present in individuals <20 years of age [25]. Therefore, further study is needed to clarify the effect of age on serum S100B levels. To date, most of the research on the clinical use of S100B has been conducted in patients with neurologic disorders or some malignancies [26,27,28,29]. Regarding liver diseases, there have been a few studies reporting serum S100B as a neuromarker of HE in patients with fulminant hepatitis [16,17,30]. Especially, Toney et al. investigated that individual neuromarkers derived from specialized cell types within the brain, such as neuron-specific enolase (NSE) present in neurons, S100B present in astrocytes, and myelin basic protein (MBP) in oligodendrocytes, would be associated with the development of HE in pediatric acute liver failure [16]. They found that only S100B was associated with HE. Additionally, they showed no relationship between S100B and the severity of HE, similar to the present study. There is another study using samples from 54 subjects in the US Acute Liver Failure Study Group. In that study, Vaquero et al. found that S100B was increased in subjects with (i) stage 1–2 HE who did not progress, (ii) stage 1–2 HE who progressed to a severe encephalopathy, (iii) stage 3–4 HE who survived and (iv) stage 3–4 HE who died or required transplantation [31]. However, a significant difference in S100B level according to the severity of HE was not observed. Hepatic stellate cells, which share many functional and morphologic characteristics with glial cells, were suggested as another relevant source of serum S100B in the setting of liver failure.

Taken together, obvious astrocyte swelling and damage developed in acute liver failure may release S100B protein; thus, elevated serum S100B may be a marker suggesting severe HE. However, based on the results of the present study, the elevated serum S100B levels observed in fulminant hepatic failure may be the result of not only astrocyte injury but also liver damage itself.

HE is a typical cirrhosis-related complication and is associated with repeated readmission and, more importantly, high rates of mortality [32,33]. Although MHE is the mildest form in the HE spectrum, it is associated with low health-related quality of life (QOL) and worse survival [34,35]. Because patients with MHE have no clinically detectable neurological-psychiatric abnormalities, neuropsychological or electrophysiological tests are required to diagnose MHE [36]. Under real-world conditions, these diagnostic procedures are cumbersome and time-consuming for clinicians. Therefore, the identification of serologic markers for diagnosing MHE will benefit the current clinical practice for cirrhotic patients. Unfortunately, in the present study, serum S100B levels did not show diagnostic value for MHE or overt HE. Ammonia levels also failed to show a direct correlation with the severity of HE.

However, in the present study, serum S100B levels had a significant relationship with the MELD score regardless of the presence of HE. It is uncertain how serum S100B levels are elevated in cirrhotic patients without brain injury. Cell damage followed by apoptosis or necrosis, which occurs in the liver, is presumed to be the possible source of blood S100B protein. S100B is known to play various roles through intracellular and extracellular functions [37]. In our previous experimental study, we found that hepatic stellate cells, the key players in liver fibrosis, are activated by recombinant S100B protein treatment [12]. Recently, another study has elucidated the role of S100B in hepatocellular carcinoma (HCC) [38]. Yan et al. showed that S100B expression is correlated with hypoxia and the immune response in a human HCC cell line. Therefore, S100B may be involved in various liver diseases. To clarify the mechanism of S100B in liver diseases, further research is needed.

The present study indicated that the combination of a high MELD score and S100B levels, high ammonia levels, and low cholesterol levels were associated with 1-year mortality in cirrhotic patients. Recently, Tranah et al. reported that plasma ammonia is an independent predictor of hospitalization with liver-related complications and mortality in clinically stable outpatients with cirrhosis through a prospective cohort study [39]. As a gut-derived neurotoxin, hyperammonemia resulting from impaired metabolism in chronic liver diseases leads to cerebral edema and intracranial pressure elevation [40,41]. Additionally, ammonia is known to be related to the pathogenesis of liver-related complications, including liver cell damage, immune dysfunction, sarcopenia, and portal hypertension [42]; thus, together with S100B, ammonia may be an interesting indicator of LC. Similar to our findings, Feng et al. reported that decreased total cholesterol is significantly associated with reduced survival in cirrhotic patients [43].

The results of the present study should be interpreted considering its limitations. First, the sample size was relatively small, but this study was conducted prospectively and used proper neuropsychological tests to discriminate MHE and overt HE. In the present data, the prevalence of MHE and mortality-related factors in cirrhotic patients were similar to those of recently published studies, reflecting the adequacy of the present study. The sample size of the non-HE group was also too small to perform survival analysis. Nevertheless, our results showed that S100B is worthy of further study in cirrhotic patients without any types of HE. Second, the value of serum S100B in the age-matched healthy control group was not investigated. Furthermore, seasonal variation (summer/winter) or genetic factors (S100B single nucleotide polymorphism rs9722 and S100B haplotype T-G-G-A) could affect the levels of serum S100B [44,45]. Therefore, in liver disease, subsequent research on the clinical value of S100B should include a normal control group and be designed considering the source of variation affecting the levels of serum S100B concentrations. Lastly, in the present study, MHE was diagnosed using MMSE and PHES. Recently, two abnormal tests among neuropsychological assessments (i.e., PHES, critical flicker frequency test, inhibitory control test, and Stroop test et al.) have been required to diagnose MHE [46]. Unfortunately, the present study was designed in early 2013. At that time, no agreement existed as to what combination of tests should be carried out to confirm MHE. We followed a general approach to the diagnosis of MHE based on Ferenci et al. [47]. Therefore, subsequent research using more various neuropsychological assessment tools is necessary.

In summary, this is a negative study showing that serum S100B is not useful for MHE diagnosis; however, the present study found that serum S100B levels can be affected by age, serum bilirubin, and creatinine in cirrhotic patients, and associated with MELD score. Additionally, serum S100B levels showed the possibility of predicting 1-year mortality in cirrhotic patients. In the future, basic and clinical studies are necessary to identify the effect of liver dysfunction on serum S100B levels in various liver diseases.

## Figures and Tables

**Figure 1 diagnostics-13-00333-f001:**
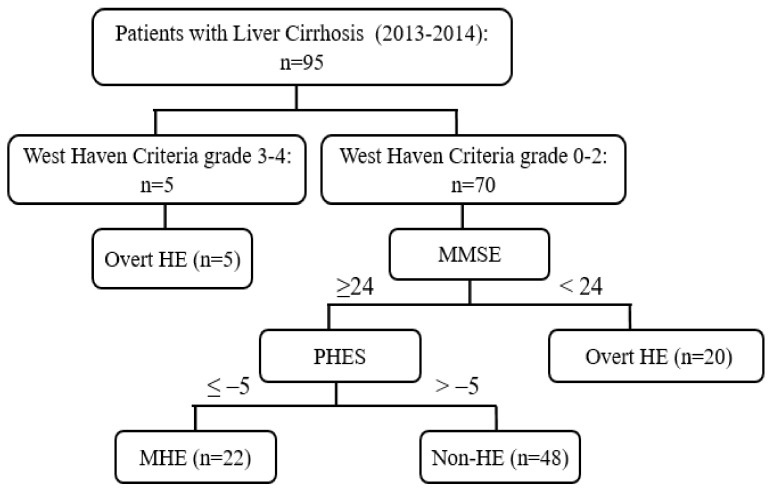
Flowchart of study patients. HE, hepatic encephalopathy; MMSE, mini-mental state examination; PHES, psychometric hepatic encephalopathy score; MHE, minimal hepatic encephalopathy.

**Figure 2 diagnostics-13-00333-f002:**
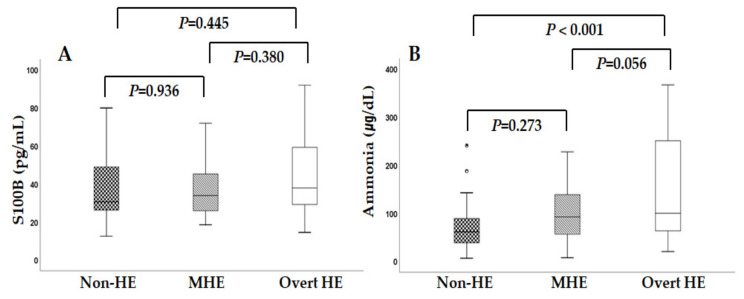
Comparision of serum S100B and ammonialevels according to the severity of HE; (**A**): There was no significance differences in serum S100B among the three groups. (**B**): Serum ammonia levels showed significant difference only between the overt HE group and the non-HE group.

**Figure 3 diagnostics-13-00333-f003:**
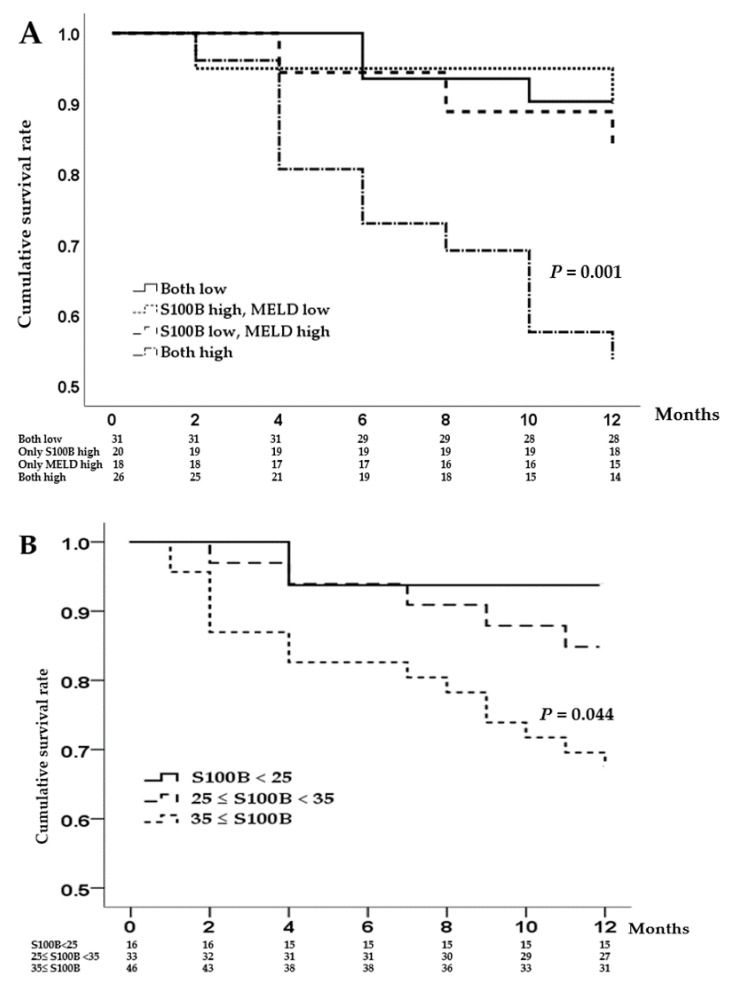
Kaplan–Meier curve displaying the estimated survival probability for the different groups of patients; (**A**): When the S100B level and MELD score were high, 1-year mortality rates were significantly higher than other groups. (**B**): Even without the MELD score, the higher the S100B levels, the higher mortality rates were observed.

**Figure 4 diagnostics-13-00333-f004:**
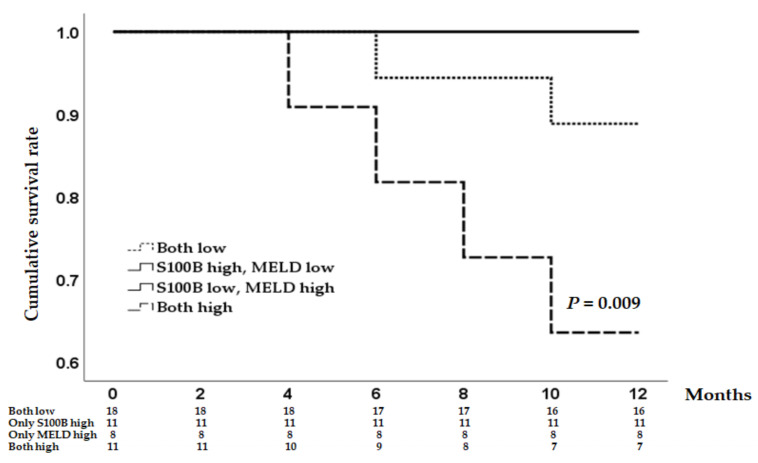
Kaplan–Meier curve displaying the estimated survival probability for the four different groups of patients without any types of hepatic encephalopathy.

**Table 1 diagnostics-13-00333-t001:** West Haven criteria.

Grade	Features
0	No abnormality detected
I	Trivial lack of awareness, euphoria, or anxiety, shortened attention span, impairment of addition or subtraction
II	Lethargy or apathy, disorientation for time, obvious personality change, inappropriate behavior
III	Somnolence to semistupor, responsive to stimuli, confused, gross disorientation, bizarre behavior
IV	Coma, unable to test mental state

**Table 2 diagnostics-13-00333-t002:** Comparison of clinical characteristics of the study groups.

Variables	Non HE (n = 48)	MHE (n = 22)	Overt HE (n = 25)	*p*
Age (years)	49.6 ± 7.1	48.9 ± 7.3	62.3 ± 11.6	<0.001
Sex (male, %)	36 (75)	17 (77.3)	17 (68)	0.739
BMI (kg/m^2^)	23.6 ± 3.3	23.6 ± 3.2	24.3 ± 3.9	0.674
Cirrhosis etiology, n	33/11/2/2	16/3/1/2	14/5/2/4	0.647
(Alcohol/HBV/HCV/Other, %)	(68.8/22.8/4.2/4.2)	(72.8/13.6/4.5/9.1)	(56/20/8/16)	
WBC (×10^3^/µL)	5.2 ± 2.5	5.4 ± 2.1	5.6 ± 3.2	0.831
Hemoglobin (g/dL)	10.3 ± 1.5	10.7 ± 2.1	11.1 ± 2.2	0.198
Platelet (×10^3^/µL)	115.4 ± 75.3	119.3 ± 81.6	88.7 ± 39.2	0.225
Cholesterol (mg/dL)	130.2 ± 49.1	120.8 ± 30.7	117.8 ± 62.3	0.549
AST (IU/L)	103.6 ± 134.6	73.3 ± 72.3	107.0 ± 296.3	0.776
ALT (IU/L)	50.1 ± 87.0	36.5 ± 57.7	44.4 ± 107.4	0.830
GGT (IU/L)	188.8 ± 233.8	177.3 ± 181.4	138.5 ± 221.1	0.647
ALP (U/L)	216.3 ± 171.6	218.2 ± 154.6	144.1 ± 93.6	0.123
Uric acid (mg/dL)	4.8 ± 2.1	5.1 ± 2.1	5.9 ± 2.9	0.177
LD (IU/L)	419.8 ± 174.2	434.3 ± 186.9	500.8 ± 643.2	0.658
INR	1.6 ± 0.6	1.6 ± 0.4	1.6 ± 0.7	0.861
Total Bilirubin (mg/dL)	3.5 ± 3.6	4.5 ± 5.2	3.6 ± 4.8	0.673
Albumin (g/dL)	2.9 ± 0.6	2.9 ± 0.5	2.9 ± 0.5	0.992
Creatinine (mg/dL)	0.8 ± 0.5	0.8 ± 0.2	0.9 ± 0.4	0.482
CRP (mg/L)	5.4 ± 5.1	9.6 ± 14.6	12.4 ± 13.9	0.057
Ammonia (µg/dL)	69.5 ± 45.7	100.3 ± 57.2	153.32 ± 112.6	<0.001
CTP (A/B/C, n)	8/25/15	3/9/10	3/13/9	0.819
MELD	11.9 ± 7.1	13.8 ± 5.0	13.5 ± 8.2	0.501
S100B (pg/mL)	38.1 ± 17.2	36.5 ± 13.9	43.5 ± 19.7	0.318

HE, hepatic encephalopathy; MHE, minimal hepatic encephalopathy; BMI, body mass index; HBV, hepatitis B virus; HCV, hepatitis C virus; WBC, white blood cell; AST, aspartate aminotransferase; ALT, alanine aminotransferase; GGT, gamma-glutamyl transferase; ALP, alkaline phosphatase; LD, lactate dehydrogenase; INR, International Normalized Ratio; CRP, C-reactive protein; CTP, Child-Turcotte-Pugh; MELD, Model for End-Stage Liver Disease.

**Table 3 diagnostics-13-00333-t003:** Clinical characteristics of the patients according to serum S100B levels.

Parameters	S100B Group			*p*
	S100B < 25	25≤ S100B <35	35≤ S100B	
	n = 16	n = 33	n = 46	
Age (years)	46.6 ± 9.8	53.5 ± 8.2	54.4 ± 10.2	0.027
Sex (male, %)	16 (100)	23 (69.7)	31 (67.4)	0.031
BMI (kg/m^2^)	25.1 ± 4.3	22.6 ± 2.9	24.2 ± 3.3	0.026
Cirrhosis etiology, n	11/3/0/2	18/11/3/1	34/5/2/5	0.239
(Alcohol/HBV/HCV/Other, %)	(68.8/18.8/0/12.5)	(54.5/33.3/9.1/3)	(73.9/10.9/4.3/10.9)	
WBC (×10^3^/µL)	5.3 ± 2.9	5.6 ± 2.7	5.2 ± 2.5	0.820
Hemoglobin (g/dL)	10.2 ± 1.8	10.9 ± 1.8	10.5 ± 1.9	0.486
Platelet (×10^3^/µL)	140.1 ± 96.9	122.7 ± 81.4	89.9 ± 38.1	0.015
Cholesterol (mg/dL)	130.4 ± 36.1	134.7 ± 54.2	115.7 ± 49.1	0.215
AST (IU/L)	48.3 ± 25.7	84.8 ± 115.8	123.7 ± 238.9	0.320
ALT (IU/L)	23.6 ± 18.0	39.9 ± 56.7	57.1 ± 113.8	0.374
GGT (IU/L)	108.9 ± 91.1	217.1 ± 289.1	163.5 ± 185.2	0.247
ALP (U/L)	150.8 ± 90.1	241.9 ± 197.9	182.4 ± 125.4	0.093
Uric acid (mg/dL)	4.9 ± 1.9	4.9 ± 1.9	5.3 ± 2.9	0.697
LD (IU/L)	318.8 ± 101.7	457.2 ± 152.6	479.1 ± 494.0	0.302
INR	1.4 ± 0.3	1.5 ± 0.5	1.7 ± 0.6	0.120
Total Bilirubin (mg/dL)	2.4 ± 1.7	2.8 ± 2.1	4.9 ± 5.7	0.036
Albumin (g/dL)	3.0 ± 0.5	3.1 ± 0.5	2.9 ± 0.6	0.300
Creatinine (mg/dL)	0.8 ± 0.3	0.7 ± 0.2	0.9 ± 0.6	0.025
CRP (mg/L)	6.1 ± 5.6	8.3 ± 12.3	9.1 ± 11.5	0.671
Ammonia (µg/dL)	95.7 ± 92.6	101.4 ± 80.1	97.3 ± 73.9	0.300
CTP (A/B/C, n)	3/7/6	4/21/8	7/19/20	0.354
MELD	9.6 ± 5.0	10.5 ± 5.5	15.6 ± 7.5	0.001

BMI, body mass index; HBV, hepatitis B virus; HCV, hepatitis C virus; WBC, white blood cell; AST, aspartate aminotransferase; ALT, alanine aminotransferase; GGT, gamma-glutamyl transferase; ALP, alkaline phosphatase; LD, lactate dehydrogenase; INR, International Normalized Ratio; CRP, C-reactive protein; CTP, Child-Turcotte-Pugh; MELD, Model for End-Stage Liver Disease.

**Table 4 diagnostics-13-00333-t004:** Relationships of S100B with clinical and biochemical parameters.

Parameters	S100B	
	Correlation Coefficient	*p*
Age	0.223	0.030
WBC	0.032	0.761
Platelet	−0.229	0.026
Cholesterol	−0.208	0.043
Uric acid	0.271	0.008
LD	0.239	0.020
ALT	0.239	0.020
INR	0.207	0.044
Total bilirubin	0.262	0.010
Albumin	−0.210	0.041
Creatinine	0.320	0.002
CRP	0.182	0.088
Ammonia	0.096	0.380
CTP score	0.199	0.054
MELD	0.385	<0.001

WBC, white blood cell; ALT, alanine aminotransferase; LD, lactate dehydrogenase; INR, International Normalized Ratio; CRP, C-reactive protein; CTP, Child-Turcotte-Pugh; MELD, Model for End-Stage Liver Disease.

**Table 5 diagnostics-13-00333-t005:** Multivariate linear regression analysis.

Variables	β	SE	t	95%CI	*p*
Age	0.354	0.162	2.183	0.032–0.676	0.032
Platelet	−7.154	7.262	−0.985	−21.587–7.280	0.327
Cholesterol	−0.022	0.037	−0.612	−0.095–0.051	0.549
Uric acid	1.098	0.752	1.459	−0.397–2.592	0.148
LD	0.000	0.006	0.053	−0.012–0.013	0.958
ALT	0.026	0.025	1.040	−0.024–0.076	0.301
INR	−0.647	4.593	−0.141	−9.778–8.484	0.888
Total Bilirubin	1.138	0.376	2.952	0.365–1.864	0.004
Albumin	−3.704	3.155	−1.174	−9.973–2.566	0.244
Creatinine	10.416	3.709	2.808	0.311–2.137	0.006

ALT, alanine aminotransferase; LD, lactate dehydrogenase; INR, International Normalized Ratio.

**Table 6 diagnostics-13-00333-t006:** Comparison of clinical characteristics of surviving and non-surviving patients at 1-year.

Variables	Survive (n = 74)	Deceased (n = 21)	*p*
Age (years)	51.4 ± 9.2	57.5 ± 12.4	0.046
Sex (male, %)	53 (71.6%)	17 (81.0%)	0.289
BMI (kg/m^2^)	23.9 ± 3.6	23.5 ± 2.7	0.0632
Non-HE/MHE/Overt HE	41/16/17	7/6/8	0.184
Cirrhosis etiology, n	47/16/3/8	16/3/2/0	0.412
(Alcohol/HBV/HCV/Other), %	(63.5/21.6/4.1/10.9)	(76.2/14.3/9.5/0)	
WBC (×10^3^/µL)	5.0 ± 2.2	6.6 ± 3.5	0.012
Hemoglobin (g/dL)	10.7 ± 1.8	10.3 ± 2.0	0.377
Platelet (×10^3^/µL)	114.1 ± 74.2	92.1 ± 49.2	0.203
Cholesterol (mg/dL)	133.2 ± 48.5	95.1 ± 41.1	0.001
AST (IU/L)	84.8 ± 113.9	142.1 ± 322.8	0.433
ALT (IU/L)	41.4 ± 71.2	59.6 ± 128.3	0.531
GGT (IU/L)	190.7 ± 229.6	110.3 ± 161.5	0.137
ALP (U/L)	207.1 ± 162.3	164.6 ± 109.5	0.262
Uric acid (mg/dL)	4.9 ± 2.1	5.9 ± 3.2	0.172
LD (IU/L)	408.1 ± 172.9	572.4 ± 691.9	0.293
INR	1.5 ± 0.4	2.1 ± 0.8	0.004
Total Bilirubin (mg/dL)	2.9 ± 2.8	6.8 ± 6.9	0.019
Albumin (g/dL)	3.1 ± 0.5	2.7 ± 0.5	0.008
Creatinine (mg/dL)	0.8 ± 0.4	1.1 ± 0.6	0.079
CRP (mg/L)	7.8 ± 10.4	10.0 ± 13.0	0.428
Ammonia (µg/dL)	84.5 ± 67.5	147.7 ± 94.8	0.012
S100B	37.1 ± 16.3	46.4 ± 18.9	0.027
CTP score	8.4 ± 1.8	9.9 ± 2.3	0.001
MELD	10.9 ± 5.3	19.3 ± 8.2	<0.001

BMI, body mass index; HBV, hepatitis B virus; HCV, hepatitis C virus; WBC, white blood cell; AST, aspartate aminotransferase; ALT, alanine aminotransferase; GGT, gamma-glutamyl transferase; ALP, alkaline phosphatase; LD, lactate dehydrogenase; INR, International Normalized Ratio; CRP, C-reactive protein; CTP, Child-Turcotte-Pugh; MELD, Model for End-Stage Liver Disease.

**Table 7 diagnostics-13-00333-t007:** Univariate and multivariate Cox analyses for predicting 1-year mortality.

Variables	Univariate Analysis			Multivariate Analysis		
	HR	95% CI	*p*	HR	95% CI	*p*
Age	1.058	1.008–1.110	0.021			
Low cholesterol (<97 or not)	6.256	2.186–17.904	0.001	7.105	1.832–27.553	0.005
Ammonia	1.009	1.003–1.016	0.004	1.010	1.002–1.018	0.011
CTP (C or not)	3.152	1.162–8.547	0.024			
S100B (≥35 or not)	3.468	1.209–9.943	0.021			
MELD (≥13 or not)	5.257	1.735–15.930	0.003			
S100B & MELD high	7.625	2629–22.118	<0.001	6.758	1.817–25.131	0.004

AST, aspartate aminotransferase; ALT, alanine aminotransferase; LD, lactate dehydrogenase; MELD, Model for End-Stage Liver Disease.

**Table 8 diagnostics-13-00333-t008:** Relationships between S100B and clinical parameters in the non-HE group.

Parameters	S100B	
	Correlation Coefficient	*p*
Age	0.004	0.977
WBC	0.178	0.226
Platelet	−0.161	0.275
Cholesterol	−0.112	0.448
AST	0.210	0.152
ALT	0.223	0.127
INR	0.254	0.081
Total bilirubin	0.370	0.010
Albumin	−0.144	0.328
Creatinine	0.281	0.053
CRP	0.003	0.984
Ammonia	0.125	0.424
CTP score	0.254	0.082
MELD	0.464	0.001

WBC, white blood cell; AST, aspartate aminotransferase; ALT, alanine aminotransferase; INR, International Normalized Ratio; CRP, C-reactive protein; CTP, Child-Turcotte-Pugh; MELD, Model for End-Stage Liver Disease.

## Data Availability

The data presented in this study are available on request from the corresponding author. The data are not publicly available because they are property of the Institute for Liver and Digestive Diseases, Hallym University, Republic of Korea.

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
