# Peer review of "Serum S100B Levels in Patients with Liver Cirrhosis and Hepatic Encephalopathy"

_diagnostics, 2023, doi:10.3390/diagnostics13030333_

Round 1
Reviewer 1 Report
This manuscript entitled “ New Clinical Value of Serum S100B in Liver Cirrhosis” led by MO-Jong Kim et al., aimed to investigate the diagnostic ability of serum S100B to discriminate the grades of hepatic encephalopathy (HE) among liver cirrhotic patients. Authors also investigated whether serum S100B levels can be 32 used to predict 1-year mortality in cirrhotic patients. The study used only 95 cirrhotic patients were consecutively enrolled and divided into the following three groups: 1) without any types of HEs; ii) with minimal HE (MHE); and iii) with overt HE. The diagnosis of MHE was made with the previously established method of Mini-Mental State Examination (MMSE) and Psychometric Hepatic Encephalopathy Score (PHES) and conclude that that serum S100B levels are associated with the MELD score regardless of the presence of HE. The article is interesting and expands the previous studies with increased sample number on S100B as a diagnostic marker for HE and MHE. The following comments are written to improve the present study
Comments:
a. S100B protein has been shown to change during summer and winter seasons as a source of variation, should have been incorporated to analyze in the present study subjects. Ref: https://doi.org/10.1016/j.jpsychires.2013.03.001
b. S100B is also shown to be elevated in acute fulminant hepatitis and acute on chronic liver failure patients and were shown to be unrelated to survival ref: 10.1053/jlts.2001.28742. This questions the validity of diagnostic criteria only for MHE
c. Introduction suggest that liver might be possible source of S100B. This is in contrast with the literature that decreased S100B were reported in cirrhotic livers.
d. Introduction second paragraph: To date S100B is not studied in liver cirrhosis is an oversight, should be corrected
e. S100B is altered in MHE, therefore rationale for the study should be restated towards liver cirrhosis and survival relationship in the title, abstract and rest of the manuscript
f. Age of the deceased subjects is an important aspect of survival, also how long these patients carry liver cirrhosis is a question unanswered or discussed in the manuscript
g. Consider changing the title to include S100B and MELD score, because S100B high expressing subjects among liver cirrhosis showed best survival with low MELD scores
h. Sample number in Non-HE groups is too low for a clinical study to do survival and should be included in the limitations
i. Figure 4 again proves that high S100B is associated with survival among non-HE groups
j. Vaquero et al., 2003 showed S100B is correlated with HE should be included in the discussion
k. Toney et. Al., 2019 (ref 16 in this manuscript) also showed no relationship of S100B with severity of HE, similar to the present study and should be included in the discussion in detail.
Minor comments:
l. Tables: Consider bold font for significantly different values
m. Deceased subjects is a better word to use, instead of ‘dead’
n. Figure 4 has same line for both high and low S100B or MELD
Reviewer 2 Report
The authors aimed to study S100B as a brain injury marker in liver disease.
The authors claim that this is the first study (abstract), however, this is not the case as several publications are available on pubmed. Multiple parts of the paper need to be revised to adapt to the current existing knowledge.
Title does not match the message of the paper and should be revised and include the main aim/message of the work.
The authors make an evaluation of the MHE based on the thow test MMSE and PHES. Unfortunately no flicker testing or other objective parameters that have become a standard of care have not been used.
The correlation of S100B with a MELD score is likely explained by the kidney function and should be evaluated in more detail.
Explanation on the study design: was the main point of the study to evaluate S100B or was the study performed using the retrospectively collected specimens (8 years old). What is the stability of the S100B? How were the samples stored? Is there any potential impact to expect? The vials were collected in 2013 and stored till 2020-2021 and have not been re-tawed or used before, right (2.3)? This is crucial as the authors state that the study was a prospective study.
One of the biggest challenges to state on the use of the biomarker is appropriate matching. In particular using the brain biomarker, one would expect aging as a potential co-factor. This is the case of the study where the overt HE have higher age compared to other groups.
This is a negative study showing that S100B is not useful for MHE diagnosis. I am quite concerned about the main message of the paper while looking at table 2. The Ammonia levels show as expected significant differences, while S100B shows basically limited variation and no change between the main groups. This means the primary aim of the study has a negative output, right? This is important to make the conclusions in abstract and the main paper.
The analysis of the value of S100B in prognosis should include the non MELD comparison. This is not the case at present. So the authors need to provide 2-3 groupings depending on the level of S100B and show the impact on the prognosis independently on MELD. Whether S100B adds additional value as MELD alone is not shown.
Creatinine/renal function is likely one of the key factors that may impact the stability and clearance of S100B. This should be carefully evaluated. At least the data in Table 4 demonstrate the biggest impact. Why this relationship was not considered in Table 5 is not clear.
Round 2
Reviewer 1 Report
Authors have significantly improved their manuscript based on both the reviewers comments.
Author Response
We appreciate your thoughtful comments. We hope that the paper will now be considered for publication.
Thank you very much.
Reviewer 2 Report
The paper has been revised to address the comment of reviewers.
Some of the comments have been addressed, while few others are not yet addressed. Please, allow me to make the comments as direct and precise as possible to overcome misunderstanding.
The title may be more objective for instance: “Serum S100B Levels in Patients with liver cirrhosis and hepatic encephalopathy”. If the authors consider the prior version then wording “correlated” should be exchanged to “associated with”. Hepatic encephalopathy is actually the primary aim of the study and it was not the aim of the work to compare the association-level. The primary aim of the study actually fails to show any value of S100B in HE and correctly may be part of the title..
The authors claim the clinical value of S100B. However, I did not see results of any additional benefit of S100B in comparison to MELD-Score. Unless clearly shown, the statement needs revision (Abstract/paper). The conclusions of the work are: 1. S100B has no role in HE diagnostic. 2. Association with MELD but what is the clinical relevance? Treatment? What difference would it make?
I still do not understand why the authors avoid implementation of creatinine in multivariate analysis. To my view it is important to understand what are the factors that may be associated with changes in S100B levels. If the authors believe that Bilirubin might be also helpful to include, this would be also welcome.
Figure 2. What specific post test has been used following ANOVA testing? The information needs to be implemented. T-test would be inappropriate in this case.
The table 3 is quite helpful, however, I would strongly recommend for homogeneity to use the same pattern as survival data analysis, meaning that grouping into 3 would be appropriate here (the same as in Figure 3).
Author Response
We appreciate your thoughtful and helpful comments. We hope that the paper will now be considered for publication.
Please see the attachment.

Round 3
Reviewer 2 Report
The comment have been adequately addressed in the revised version.